# Changes in the Microbial Community in Soybean Plots Treated with Biochar and Poultry Litter

**Rosalie B. Calderon** [1,2], **Changyoon Jeong** [3], **Hyun-Hwoi Ku** [3,4], **Lyndon M. Coghill** [5,6], **Young Jeong Ju** [5], **Nayong Kim** [5] and **Jong Hyun Ham** [1,*]

1    Department of Plant Pathology and Crop Physiology, Louisiana State University Agricultural Center, Baton Rouge, LA 70803, USA; RCalderon@agcenter.lsu.edu
2    Department of Plant Pathology, College of Agriculture, Benguet State University, La Trinidad 2601, Philippines
3    Red River Research Station, Louisiana State University Agricultural Center, Bossier City, LA 71112, USA; CJeong@agcenter.lsu.edu (C.J.); sowe2011@gmail.com (H.-H.K.)
4    School of Applied Science in Natural Resources and Environment, Hankyong National University, Anseong 456-749, Korea
5    Center for Computation and Technology, Louisiana State University, Baton Rouge, LA 70803, USA; lcoghill@missouri.edu (L.M.C.); dudwjd93@gmail.com (Y.J.J.); nykim@cct.lsu.edu (N.K.)
6    Office of Research, Informatics Research Core, University of Missouri, Columbia, MO 65211, USA
*    Correspondence: jham@agcenter.lsu.edu; Tel.: +1-(225)-578-6798

**Abstract:** The application of organic materials that promote beneficial microbial activity is vital to maintaining soil health and crop productivity. We investigated the effect on the soil microbiome of applying biochar (BC), poultry litter (PL), and a combination of biochar and poultry litter (BC/PL) in soybean cultivation at the Red River Research Station (Bossier City, LA, USA). We characterized the microbial profiles, community structure, and co-occurrence network from sequencing data to infer microbial interactions in the soil samples collected in the first and second years of each soil treatment (2016 and 2017, respectively). Our results showed that soil treatments with BC, PL, and a combination of both moderately changed the microbial community composition and structure. In particular, genera significantly affected by the different soil treatments were identified via differential abundance analysis. In addition, canonical correspondence analysis revealed that soil chemical properties, total N in the first year, and total C and pH in the second year influenced the community variability. The differentially enriched bacterial ASVs and co-occurring taxa were linked to nutrient cycling. This study provides insights into the impact of soil carbon amendment on the soil microbiome, a process which favors beneficial bacteria and promotes soybean growth.

**Keywords:** soil microbiome; soybean; biochar; soil amendment; microbial community network

## 1. Introduction

Intensive farming practices, including the usage of pesticides and synthetic fertilizers, hamper sustainable agriculture methods which provide a stable supply of food and feed without significant environmental damage [1]. The soybean (*Glycine max.* L.) is the most valuable legume crop, ranking as the ninth most-produced crop in the world [2]. As a leguminous plant, the soybean can acquire nitrogen through the N-fixation by symbiotic and asymbiotic bacteria in nodules and root surfaces [3–5].

Soil microbial communities are vital in the ecosystem, playing key roles in processes such as nutrient and soil carbon cycles [6–8]. Likewise, modifications in the soil microbial community through management practices such as the addition of soil amendments subsequently impact their assemblage and composition in the plant rhizospheres [6,8–10]. In this regard, improving knowledge on the capacity of the soil microbiome is crucial for sustainable crop management, and will aid in understanding the influence of the soil

microbiome on soil nutrients for plants, suppressing diseases, and increasing tolerance to abiotic stresses.

Soil carbon amendments include a diverse array of inputs from animal litter, solid waste, and compost. One of these is biochar, a pyrogenic carbon widely distributed in the United States, Europe, Asia, and Africa [11–14]. Biochar is reported to be rich in labile carbon fractions, which improve soil fertility, induce systemic resistance in plants against soil-borne pathogens, and enhance soil biomass and bioactivity [1,14,15]. The physicochemical properties of biochar, including high C content and high adsorption characteristics, increase pH in acidic soils and improve nutrient retention [16]. Additionally, the reduction of soil tensile strength by biochar and the large internal surface area of biochar have been considered to modify soil microbial abundance and diversity [14]. Meanwhile, poultry litter could be a potent alternative nutrient source in order to increase soil fertility and enhance crop yield [17]. However, an effective strategy to prevent rapid nutrient loss is required to address one of the critical issues hindering the environmentally safe application of poultry litter to agricultural land. Most of the nutrient losses from agricultural lands are through soil erosion from irrigated agriculture or runoff after rain events [18]. Recently, biochar has been proposed as a possible soil amendment to sequester carbon (C) in soils and to prevent rapid nutrient loss from crop fields [19,20]. The reported cation exchange capacity (CEC) values range from 83.0 to 113.7 $cmol_c$ $kg^{-1}$ for sugarcane leaf biochar and from 48.0 to 68.0 $cmol_c$ $kg^{-1}$ for rice straw biochar [21].

Recent studies on the soybean microbiome revealed the presence of both rhizobial and non-rhizobial bacteria in the nodules [22,23]. At different soybean growth stages in agricultural fields, it was found that there were varying bacterial communities in the root-associated and rhizosphere microbiomes [4,24]. Furthermore, a soybean biochar study in greenhouse conditions revealed a combined effect on growth and a microbiome shift in root-associated bacteria [25]. In addition, some field studies revealed the influence of biochar application on the soil microbial community due to the alteration of physicochemical properties, as well as an increase in nutrients, water-holding capacity, and aeration in the soil systems [16,26–28]. Biochar application as a form of soil amendment also increased recalcitrant organic carbon as compared to the application of compost or animal manure, and as such can be used to control $CO_2$ evolution [16].

Synergistic effects have been reported on crop yield and growth under the application of biochar combined with inorganic fertilizers [29–31]. Moreover, the application of both biochar and fertilizer significantly affected the phospholipid fatty acid (PLFA) soil microbial community structure as compared to that of biochar alone. Additionally, the combined application of biochar and manure or the sole application of manure significantly influenced the PLFA soil microbial community structure and soil enzyme activity in short-term studies [32–34].

Only a limited number of long-term field studies have been documented on the influence of their combined application on the changes in soil microbial abundance and community composition. In this study, we analyzed the soil microbiomes in soybean field using high-throughput sequencing methods with 16S amplicon metagenomics to characterize the microbial community profiles, structures, and co-occurrence network in the conditions of different soil amendments. Specifically, we characterized the soil microbial community and the differentially abundant taxa from a soybean field treated with biochar, poultry litter, or a combination of both, along with the plots of an untreated control. Moreover, soil properties such as pH, total nitrogen, total carbon, and the C:N ratio were determined in association with the microbial structure. This research will elucidate the effect of soil carbon amendments on the soil microbiome and the potential utilization of biochar and poultry litter as an eco-smart strategy to prevent nutrient loss and favor beneficial bacteria for sustainable crop production.

## 2. Materials and Methods

### 2.1. Soil Treatments, Soybean Culture Conditions, and Collection of Soil Samples

The field experiment for this study was conducted at the Red River Research Station, LSU AgCenter (Bossier City, LA, USA; 32°24′54.31″ N, 93°37′48.30″ W), whereas the experimental plot was laid out according to a split-plot design with 4 replications. The 4 soil treatments in this study were untreated control (UT), biochar (BC) at 11.2 Mg ha$^{-1}$, poultry litter (PL) at 0.8 Mg ha$^{-1}$ (2.8% as P) dry weight, and a combination of biochar and poultry litter (BC/PL) based on the LSU AgCenter recommendation [35]. The treatments were applied yearly for 2 cropping seasons (2016 and 2017). The applied biochar was made of yellow pine waste created by fast pyrolysis at 500 °C (Waste To Energy, Inc., Slocomb, AL, USA). Poultry litter was obtained from the broiler house at the Hill Farm Research Station, LSU AgCenter (Homer, LA, USA; 32°45′12.9996″ N, −93°04′33.6972″ W). The soil was plowed to a depth of 30 cm in March. Biochar and poultry litter were then applied by broadcast on top of the soybean plots and incorporated with a rake to a depth of 10 cm. Basal fertilizers were calcium superphosphate and potassium sulfate at rates of 120 kg $P_2O_5$ ha$^{-1}$ and 100 kg $K_2O$ ha$^{-1}$. There was no nitrogen fertilization because the field was continuously planted with leguminous plants. Soybean plants were planted in May 2016 and 2017 and harvested in October 2016 and 2017. The yield data included the total weight of soybean seeds harvested per plot/replicate per treatment in kg/ha. The variations in soybean growth and yield were determined using one-way ANOVA.

Soil samples were collected randomly from every experimental plot. A soil composite composed of 10 cores was collected from each replicate plot using a soil core sampler at a depth of 15 cm with a 2.5 cm diameter 3 days before harvest to confirm the fully matured microbial community changes in the bulk soil. Each soil sample per replicate was divided for physicochemical analyses and DNA extraction.

### 2.2. Soil Physicochemical Analyses

Soil pH was measured using a pH meter at a 1:1 soil:water ratio [36]. The total C and N contents in the soil were measured by dry combustion using a Thermo Finnigan FLASH EA 1112 CN analyzer (CE Elantech, Inc., Lakewo1od, NJ, USA) at the water quality laboratory at the Red River Research Station, LSU AgCenter [36]. Mehlich-3 extractable nutrients (P, K, Ca, Mg, and S) were determined with induced coupled plasma atomic emission spectroscopy (ICP-AES) (Spectro CIROSCCD, Mahwah, NJ, USA) from the Soil Testing Laboratory, LSU AgCenter [36]. Bulk density was measured by collecting multiple cores of known volume from different irrigation schemes [37].

### 2.3. Biochar and Poultry Litter Analysis

The chemical characteristics of the biochar and the poultry litter used in this study (Table 1) were analyzed with the following protocols. The pH of biochar samples was measured using a pH meter at a 1:5 solid:water ratio. The ash content of the biochar was determined by overnight combustion at 750 °C in a muffle furnace. The total C and total N were analyzed with the same method as soil sample analysis. The macro- and micro-element contents in biochar and poultry litter were analyzed from the Soil Testing Laboratory at the LSU AgCenter.

**Table 1.** Selected chemical properties of biochar and poultry litter applied to the experimental plots.

| Property | Units | Biochar | Poultry Litter |
|----------|-------|---------|----------------|
| pH | | 8.9 | 7.34 |
| EC | dS m$^{-1}$ | 0.8 | 13.5 |
| Ash | % | 15 | N/A * |
| Total C | % | 69.1 | 27.2 |
| Total N | % | 0.28 | 3.86 |
| Organic matter | % | N/A | 24.54 |
| Ca | mg kg$^{-1}$ | 108 | 4322 |
| Cu | mg kg$^{-1}$ | ND ** | 101 |
| Mg | mg kg$^{-1}$ | 25 | 3977 |
| P | mg kg$^{-1}$ | 38 | 1393 |
| K | mg kg$^{-1}$ | 59 | 4905 |
| Na | mg kg$^{-1}$ | 130 | 814 |
| S | mg kg$^{-1}$ | 20 | 2922 |
| Zn | mg kg$^{-1}$ | ND | 641 |

* N/A = not applicable. ** ND = not detected.

## 2.4. DNA Sample Processing

Total genomic DNA was extracted from 0.25 g of soil using the PowerSoil® DNA Isolation Kit (Qiagen, Germantown, MD, USA) following the manufacturer's instructions. The extracted DNA samples were analyzed for quality and quantity using NanoDrop 1000 (Thermo Scientific, Wilmington, DE, USA) and were verified with agarose gel electrophoresis. The DNA samples were sent to Genomics Research Laboratory of the Biocomplexity Institute of Virginia Tech (Blacksburg, VA, USA) for 2 × 250 bp paired-end sequencing following the 16S Illumina Amplicon Protocol from the Earth Microbiome Project (https://earthmicrobiome.org/protocols-and-standards/16s/ accessed on 1 March 2017). The standard library preparation protocol as earlier described was followed [38]. Briefly, the V4 region of the 16S small subunit rRNA gene was amplified using the universal primer set, 515F (GTGYCAGCMGCCGCGGTAA) and 806R (GGACTACNVGGGTWTCTAAT). The PCR reaction mixture included 13.0 μL of PCR-grade water, 1.0 μL of template DNA, 10 μM of each primer, and 10.0 μL of 5PRIME HotMasterMix (2×) (Quantabio, Beverly, MA, USA). Samples were amplified in duplicate under the following thermocycler conditions: 94 °C for 3 min for initial denaturing, then 35 cycles of 94 °C for 45 s, 50 °C for 60 s, and 72 °C for 90 s. A final elongation step occurred at 72 °C for 10 min followed by a hold at 4 °C. After pooling duplicates, all amplicons were visualized on a 2% agarose gel and quantitated on a Qubit fluorometer (FisherScientific, Hampton, NH, USA). Normalization was performed based on Qubit results and amplicons were pooled. The pool was quantitated by qubit and tapestation assays to determine size (380 bp). The pool was then quantitated using qPCR. Amplicons were loaded at 9.5 pM along with a 25% Phix spike and sequenced using the MiSeq v2 500-cycle kit on the Miseq platform (Illumina, Inc., San Diego, CA, USA). Quality scores were within Illumina specifications. The Illumina MiSeq sequencing generated raw reads of 6,193,760 and 5,470,755 in 2016 and 2017, respectively (Table 2). Individual samples ranged from 250,643 to 478,815 sequences per sample (Table 2).

**Table 2.** Sequence statistics: FastQC data from the sequences in 2016 and 2017.

| Raw Data | 2016 | 2017 |
|----------|------|------|
| File type | Conventional base calls | Conventional base calls |
| Encoding | Sanger/Illumina 1.9 | Sanger/Illumina 1.9 |
| Total sequence | 6,193,760 for 16 samples | 5,470,755 for 16 samples |
| Sequence length | 236–250 (Read 1); 43–250 (Read 2) | 236–250 (Read 1); 44–250 (Read 2) |
| Sequence flag as poor quality | 0 | 0 |
| Guanine/cytosine content (GC%) | 56–57 | 54–55 |
| Sequence range (total sequence per sample) | 309,916–478,815 | 250,643–471,980 |

### 2.5. Microbial Community Analysis

The microbial communities were analyzed with the Quantitative Insights into Microbial Ecology (QIIME 2 2019.4) pipeline [39]. Demultiplexed paired-end fastq files and a metadata mapping file were used as input files. Individual samples ranged from 250,643 to 478,815 sequences per sample, with a sequence length of up to 250 bp (paired-end sequences, $2 \times 250$ bp), which were the V4 region of the 16S rRNA genes of the prokaryotic community in each sample (Table 2). The Divisive Amplicon Denoising Algorithm (DADA2) in QIIME2 was used for sequence correction, removal of chimeras, and denoising. The sequences were trimmed using the –p-trim-left-f/r and –p-trunc-len-f/r function. The forward reads of all the sequences were truncated to 245 bases and the reverse reads at 128 bases and 122 bases for 2016 and 2017, respectively, based on the quality scores and the minimum length identified during subsampling from all the sequences. Paired sequences reads were joined and quality filtered using the paired-end DADA2 pipeline [40]. After quality filtering, we finally obtained 2,572,488 and 3,315,244 unique sequences for the samples from 2016 and 2017, respectively. Taxonomy was assigned to Amplicon Sequence Variants (ASVs) using the q2-feature-classifier [39] against the SILVA 128 database and BLAST+ consensus taxonomy classifier [41] directly after quality filtering and building of feature table and feature data. The unique sequences were aligned to a total of 51 phyla, 254 classes, 576 orders, 929 families, and 1763 genera (Table S1). All ASVs were aligned with MAFFT [42] and used to construct a phylogeny with fasttree2 [43].

In comparing microbial communities in 2016 and 2017, the resulting feature tables and representative sequences in each year were merged using feature-table merge and feature-table merge-sequence QIIME2 plugins for diversity analyses. The 2 years of sequencing data were merged for a direct comparison. Although differences in observed sample composition could arise due to different times of sequencing runs even with the same amplification protocol, the differences attributed to technical variation are expected to be relatively small compared to the biological variation, as previously reported [44,45].

The "q2-diversity" plugin was used to calculate the alpha diversity metrics and beta diversity metrics after samples were rarefied at an even sampling depth. The rarefaction curves showed a nice saturation pattern, indicating that the ASVs detected from this DNA sequencing represent most of the microbial community in each sample (Figure S1). Diversity and community richness were analyzed with Kruskal–Wallis non-parametric tests to compare species richness among treatments in q2 diversity metrics using observed features "observed OTUs" [46], Faith's phylogenetic diversity [47], and Shannon's diversity indices [48]. The variation in community composition was calculated using weighted and unweighted UniFrac metric [49]. Differential abundance of features was analyzed using DESEq2 package in R [50]. Nonmetric multidimensional scaling (NMDS) was conducted with the "vegan" R package using Bray–Curtis dissimilarity, unweighted Unifrac, and weighted Unifrac values among samples, which were subjected to a Wisconsin double transformation before analysis [51]. Statistical significance corresponding to differences in dispersion among groups was calculated using "permdisp" homogeneity in group dispersions and differences in means and variance using permutational analysis of variance (PERMANOVA) with 999 iterations. Other statistical analyses were conducted with JMP Pro Statistics, version 15.1.0 (SAS Institute, Cary, NC, USA).

The microbiomeSeq R package using the Adonis function was used to calculate the overall significance level and visualization of soil environmental variables in canonical correlation plots. BIOENV analysis in QIIME2 was conducted to select the best possible subsets of soil chemical variables that correlate with the community pattern using Spearman's correlation value [52].

Datasets of QIIME2 output, which include metadata of the microbiome dataset with taxonomic classifications of sequences, were further visualized using Explicit software package [53], microbiomeSeq [54] and microbiomeAnalyst [55].

## 2.6. Microbial Network Analysis

The co-occurrence network was inferred based on the Spearman's rank correlation. The ASVs with a sum relative abundance of at least 20% of all samples were subjected to a pairwise correlation [56]. A correlation between 2 ASVs was considered statistically significant at values of >0.6 for the Spearman's correlation coefficient ($r_s$) and at <0.01 for the *p*-value [57,58]. Multiple testing correction using Benjamini–Hochberg standard false discovery rate correction were used to adjust the *p*-values to reduce the chances of obtaining false-positive results [59]. The co-occurrence network is comprised of nodes and edges which represent the genus and all significant correlations identified from pairwise comparison of genera abundance. Topological properties and statistical analyses were calculated with R using the vegan [60], igraph [61], and Hmisc [62] packages. Network visualization was performed using the Gephi platform (http://gephi.github.io/ accessed on 4 July 2019), [63]). All samples were divided into groups by soil amendments.

## 3. Results

### 3.1. Soybean Growth and Yield

In the first year, no significant growth difference was observed among treatments (*p* = 0.94) (Table 3). However, in the second year, the PL and BC/PL treatments showed higher growth than BC-treated and the untreated control plots (*p* = 0.03) (Table 3). Nevertheless, there was no significant difference in yield among the treatments in both years, although the BC/PL had the highest yield among treatments over the 2 years.

**Table 3.** Effects of soil treatments on the soybean growth and yield in 2016 and 2017.

| Treatment | 2016 | | | 2017 | | |
|---|---|---|---|---|---|---|
| | Mean | Standard Deviation | Standard Error | Mean | Standard Deviation | Standard Error |
| (**A**) Plant growth (cm) | | | | | | |
| UT | 119.01 [a] | 7.04 | 3.52 | 115.38 [b] | 2.70 | 1.35 |
| BC | 119.25 [a] | 13.18 | 6.59 | 113.67 [b] | 5.27 | 2.64 |
| PL | 115.19 [a] | 17.40 | 8.70 | 121.73 [a] | 1.91 | 0.95 |
| BC/PL | 120.59 [a] | 14.59 | 7.30 | 118.87 [a,b] | 3.45 | 1.72 |
| (**B**) Yield (kg/ha) | | | | | | |
| UT | 3540.04 [a] | 49.83 | 24.91 | 3571.65 [a] | 316.43 | 158.22 |
| BC | 3549.62 [a] | 35.90 | 17.95 | 3485.06 [a] | 795.95 | 397.98 |
| PL | 3456.99 [a] | 118.99 | 59.50 | 3673.03 [a] | 327.71 | 163.86 |
| BC/PL | 4135.37 [a] | 1091.93 | 545.96 | 3945.22 [a] | 536.61 | 268.31 |

UT, untreated; BC, biochar; PL, poultry litter; BC/PL, biochar + poultry litter. Means with the same letter are not significantly different from the LSD protected Fisher's test at a 0.05% probability (*p* < 0.05).

### 3.2. Microbial Community Profiles and Structure

In terms of alpha diversity, there was no significant difference among the treatments in either of the years with respect to the ASV features species richness and Shannon's diversity based on Kruskal–Wallis non-parametric tests (Table 4 and Figure S2). However, the no-treatment (control) soil from the second year showed a significant difference as compared to the first year in terms of community richness based on phylogenetic lineage (*p* < 0.04) (Table 4B), whereas the BC- and BC/PL-treated soils in the second year were significantly different with regard to the abundance of observed features as compared the untreated soil in the first year (Table 4A). The parallel analyses conducted on species richness based on abundance and phylogenetic diversity (Faith's pd) showed measurement deviations but consistently supported that the three kinds of soil treatments (BC, PL, and BC/PL) in the second year were significantly different from the no-treatment control in the first year in terms of Shannon's diversity (*p* < 0.05) (Table 4C).

**Table 4.** Pairwise comparison of the species richness and diversity among the treatments in 2016 and 2017.

| (A) Species Richness (Observed Feature) | | | | | | | | | |
|---|---|---|---|---|---|---|---|---|---|
| **Combined Years *p* = 0.189** | | **2016 (*p* = 0.368)** | | | | **2017 (*p* = 0.355)** | | | |
| | | **UT** | **BC** | **PL** | **BC/PL** | **UT** | **BC** | **PL** | **BC/PL** |
| **2016** | UT | | 0.64 | 0.18 | 0.18 | 0.08 | 0.03 | 0.28 | 0.05 |
| | BC | 0.64 | | 0.32 | 0.32 | 0.22 | 0.16 | 0.65 | 0.18 |
| | PL | 0.18 | 0.32 | | 0.32 | 0.22 | 0.16 | 0.65 | 0.18 |
| | BC/PL | 0.18 | 0.32 | 0.32 | | 0.22 | 1.00 | 0.65 | 0.65 |
| **2017** | UT | 0.08 | 0.22 | 0.22 | 0.22 | | 1.00 | 0.24 | 1 |
| | BC | 0.03 | 0.16 | 1.00 | 1.00 | 1.00 | | 0.48 | 0.48 |
| | PL | 0.28 | 0.65 | 0.65 | 0.65 | 0.25 | 0.48 | | 0.28 |
| | BC/PL | 0.05 | 0.18 | 0.65 | 0.65 | 1.00 | 0.48 | 0.28 | |

| (B) Species Richness (Faith_pd: Faith's Phylogenetic Diversity) | | | | | | | | | |
|---|---|---|---|---|---|---|---|---|---|
| **Combined Years *p* = 0.508** | | **2016 (*p* = 0.109)** | | | | **2017 (*p* = 0.598)** | | | |
| | | **UT** | **BC** | **PL** | **BC/PL** | **UT** | **BC** | **PL** | **BC/PL** |
| **2016** | UT | | 0.24 | 0.08 | 0.24 | 0.04 | 0.08 | 0.08 | 0.08 |
| | BC | 0.24 | | 0.24 | 0.14 | 0.14 | 0.56 | 1 | 0.14 |
| | PL | 0.08 | 0.24 | | 0.38 | 0.38 | 1 | 0.24 | 0.77 |
| | BC/PL | 0.24 | 0.14 | 0.38 | | 0.24 | 1 | 1 | 0.38 |
| **2017** | UT | 0.04 | 0.14 | 0.38 | 0.24 | | 0.77 | 0.24 | 1 |
| | BC | 0.08 | 0.56 | 1 | 1 | 0.77 | | 0.38 | 0.38 |
| | PL | 0.08 | 1 | 0.24 | 1 | 0.24 | 0.38 | | 0.24 |
| | BC/PL | 0.08 | 0.14 | 0.77 | 0.38 | 1 | 0.38 | 0.24 | |

| (C) Shannon's Diversity Index | | | | | | | | | |
|---|---|---|---|---|---|---|---|---|---|
| **Combined Years *p* = 0.113** | | **2016 (*p* = 0.222)** | | | | **2017 (*p* = 0.517)** | | | |
| | | **UT** | **BC** | **PL** | **BC/PL** | **UT** | **BC** | **PL** | **BC/PL** |
| **2016** | UT | | 0.17 | 0.17 | 0.17 | 0.08 | 0.03 | 0.04 | 0.04 |
| | BC | 0.17 | | 0.31 | 0.31 | 0.22 | 0.15 | 0.17 | 0.17 |
| | PL | 0.17 | 0.31 | | 0.31 | 0.22 | 0.15 | 0.17 | 0.17 |
| | BC/PL | 0.17 | 0.31 | 0.31 | | 0.22 | 0.15 | 0.65 | 0.65 |
| **2017** | UT | 0.08 | 0.22 | 0.22 | 0.22 | | 0.64 | 0.56 | 0.56 |
| | BC | 0.03 | 0.15 | 0.15 | 0.15 | 0.64 | | 0.72 | 0.47 |
| | PL | 0.04 | 0.17 | 0.17 | 0.65 | 0.56 | 0.72 | | 0.82 |
| | BC/PL | 0.04 | 0.17 | 0.17 | 0.65 | 0.56 | 0.47 | 0.82 | |

Further analyses to determine beta diversity were performed using metrics including beta group significance (unweighted Unifrac and weighted Unifrac) and the Adonis test (Table 5). All of the metrics indicated that there was no difference among treatments in the 2 years, but there was a significant difference in each year's overall data between the 2 years (Table 5).

**Table 5.** Summary of the beta diversity metrics (*p*-value).

| Beta Diversity Metrics | Untreated vs. BC vs. PL vs. BC/PL | | | 2016 vs. 2017 |
|---|---|---|---|---|
| | **2016** | **2017** | **Combined** | |
| Beta group significance | | | | |
| Unweighted unifrac | 0.477 | 0.369 | 0.403 | 0.001 |
| Weighted unifrac | 0.146 | 0.206 | 0.287 | 0.001 |
| Adonis test | 0.020 | 0.078 | 0.419 | 0.001 |

After data filtering of uninformative ASV features with low counts and low variance (below 20% for the prevalence filter with a minimum count of 4 and below 10% variance

based on the inter-quantile range), the beta diversity patterns of the bacterial communities at the genus level were depicted in Bray–Curtis dissimilarity heatmaps and non-metric multidimensional scaling (NMDS) ordination using weighted Unifrac distance (Figure 1). Other distance methods using Bray–Curtis, unweighted Unifrac, and weighted Unifrac are presented altogether in the supplemental figure (Figure S3). We observed the tendency that the BC-treated soil showed the highest level of similarity with regard to the untreated soil in both years (Figure 1a). The relationship between the bacterial community and the treatments was not significant for the taxonomic-based distance measure using Bray–Curtis for both years. However, the weighted Unifrac distance, a phylogenetic-based distance measure which accounts for the relative abundance of taxa, showed a significant difference among treatments based on permutational multivariate analysis in the first year (PERMANOVA *F*-value: 2.0225; R-squared: 0.33582; $p < 0.02$) and a slightly substantial difference in the second year (PERMANOVA *F*-value: 1.4692; R-squared: 0.26863; $p < 0.087$). The beta dispersion using "permdisp" homogeneity in group dispersions revealed no significant difference in the spread in any treatments in the 2 years (PERMDISP *F*-value: 0.64839; $p = 0.598$ and PERMDISP *F*-value: 0.76461; $p = 0.535$, first year and second year, respectively). Thus, the difference in the microbial community composition is not ascribed to variances within the treatment but among treatments, especially in the first year.

The microbial community structures were well represented in the NMDS plots, as shown by the NMDS stress values of 0.08 and 0.11 in the first year and second year, respectively. In the first year, 33.5% variation in the microbial communities was attributed to the different treatments. There were diverging separate clusters between the BC-, PL-, and BC/PL-treated soils as compared to the untreated control in the first year (Figure 1b). In the second year, 26.8% of the variation among microbial abundances was ascribed to the different treatments, although with slight to nil significant differences ($p = 0.087$). The NMDS showed the ordination space of microbial communities in the BC- and PL-treated clusters within the ordination distance of BC/PL-treated soil, suggesting that all the microbial communities in the BC- and PL-treatments were present in the BC/PL treatments. The BC- and PL-treatments showed some portions of un-overlap ordination, suggesting that both communities showed unique microbial compositions. On the other hand, the untreated soil converged across all the treatments, which may indicate that the microbial members existing in the untreated soils could also be present in all treated soils. Thus, the results suggest that the composition of microbial community was similar, but the abundance of microbial community members varied under the different treatments. The significant PERMANOVA of the weighted Unifrac distances supported the differences in community structure among treatments because of the variation in the relative abundance of species/taxa among treatments. The correlation of microbial composition with the environmental factors could be observed by other statistical analyses (below section).

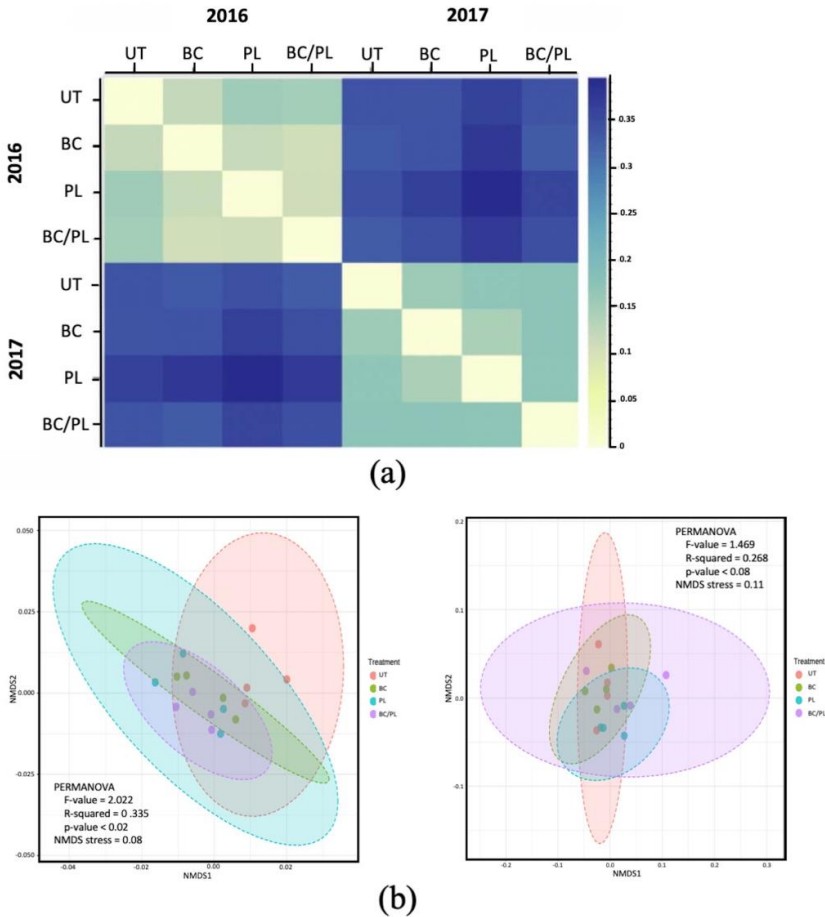

**Figure 1.** The beta diversity among soils treatments in 2016 and 2017. (**a**) A Bray-Curtis dissimilarity heatmap of the community structure and membership at the genus level among the treatments. (**b**) Non-metric multidimensional scaling (NMDS) plots with weighted Unifrac distances among soil treatments: untreated (UT), biochar (BC), poultry litter (PL), and a combination of biochar and poultry litter (BC/PL). Fields that overlap ordination space have similar community composition. Plot ellipses represent the 95% confidence regions per treatment.

### 3.3. Canonical Correspondence Analysis

Treatments influencing the soil chemical properties can change the composition of associated soil microbiomes. Although there was no significant difference in pH, total N, total C, and the C/N ratio among the treatments with one-way ANOVA (Table 6), we attempted to analyze the correlation of the environmental factors with the microbial community through multivariate statistical techniques and Bioenv and Adonis tests with the QIIME2 (Table 7). The canonical correspondence analysis (CCA) plots showed the correlation of individual soil chemical properties with the structure of the microbial community (Figure 2), and the Bioenv analysis table showed the best set of environmental factors that described the community variation in the 2 years (Table 7). Total nitrogen (R-squared: 0.124, $p = 0.039$, Spearman's $r_s = 0.741$) and the combination of total nitrogen and pH ($r_s = 0.559$) best explained the variability of microbial composition in the first year (Table 7). The BC/PL-treated soil had the highest total nitrogen values in the first year, followed by the BC- and untreated soils (Table 7). The pH (R-squared: 0.197, $p = 0.002$, $r_s = 0.358$), the combination of pH and total carbon ($r_s = 0.422$), and the combination of pH, total nitrogen, and total carbon ($r_s = 0.363$, $p = 0.031$) influenced the microbial compositions in the second year (Figure 2 and Table 7). Total carbon was higher in the BC- and the BC/PL-treated soils throughout the 2 years, especially in the BC-treated soil in the second year (Table 6).

**Table 6.** Soil chemical data in 2016 and 2017 for pH, total nitrogen, total carbon, and carbon/nitrogen ratio.

| Treatment | 2016 | | | 2017 | | |
|---|---|---|---|---|---|---|
| | Mean | Standard Deviation | Standard Error | Mean | Standard Deviation | Standard Error |
| **(A)** pH | | | | | | |
| UT | 7.68 | 0.134 | 0.067 | 6.85 | 0.244 | 0.122 |
| BC | 7.50 | 0.042 | 0.021 | 7.07 | 0.141 | 0.071 |
| PL | 7.36 | 0.226 | 0.113 | 6.87 | 0.257 | 0.129 |
| BC/PL | 7.46 | 0.271 | 0.135 | 7.18 | 0.501 | 0.250 |
| **(B)** Total nitrogen | | | | | | |
| UT | 0.045 | 0.004 | 0.002 | 0.056 [a,b] | 0.006 | 0.003 |
| BC | 0.045 | 0.006 | 0.003 | 0.063 [a] | 0.011 | 0.005 |
| PL | 0.042 | 0.008 | 0.004 | 0.036 [b] | 0.004 | 0.002 |
| BC/PL | 0.047 | 0.006 | 0.003 | 0.054 [a,b] | 0.019 | 0.009 |
| **(C)** Total carbon | | | | | | |
| UT | 0.223 | 0.042 | 0.021 | 0.381 | 0.071 | 0.036 |
| BC | 0.225 | 0.095 | 0.047 | 0.611 | 0.064 | 0.032 |
| PL | 0.171 | 0.122 | 0.061 | 0.431 | 0.200 | 0.100 |
| BC/PL | 0.373 | 0.122 | 0.061 | 0.547 | 0.139 | 0.070 |
| **(D)** C–N ratio | | | | | | |
| UT | 4.950 | 0.642 | 0.321 | 6.958 | 2.088 | 1.044 |
| BC | 4.868 | 1.461 | 0.731 | 9.953 | 1.711 | 0.855 |
| PL | 3.785 | 1.981 | 0.991 | 12.375 | 6.830 | 3.415 |
| BC/PL | 8.113 | 3.396 | 1.698 | 10.553 | 2.400 | 1.200 |

Means with the same letter are not significantly different at a 0.05% probability ($p < 0.05$).

**Table 7.** Correlation of environmental variable with the microbial composition showing the BIOENV that best explain the community variation and Adonis test in 2016 and 2017.

| Environmental Variable(s) (No. of Variables) | BIOENV [a] Spearman Coefficient ($r_s$) | Adonis Test [b] (*p*-Values) |
|---|---|---|
| **2016** | | |
| pH (1) | - | 0.145 |
| Total carbon (1) | - | 0.135 |
| Total nitrogen (1) | 0.742 | **0.039** |
| C:N ratio (1) | - | 0.294 |
| pH, total nitrogen (2) | 0.559 | 0.377 |
| pH, total nitrogen, total carbon (3) | 0.509 | 0.403 |
| pH, total nitrogen, total carbon, C:N ratio (4) | 0.472 | 1.00 |
| **2017** | | |
| pH (1) | 0.358 | **0.035** |
| Total carbon (1) | - | 0.703 |
| Total nitrogen (1) | - | 0.192 |
| C:N ratio (1) | - | 0.533 |
| pH, total carbon (2) | 0.422 | 0.165 |
| pH, total nitrogen, total carbon (3) | 0.363 | **0.031** |
| pH, total nitrogen, total carbon, C:N ratio (4) | 0.243 | 1.00 |

[a] combination of environmental variables explaining variance in soil microbiome composition; [b] analysis of variance using weighted distance matrices of individual and multiple interaction of environmental variables. Values in bold indicate statistically significant results.

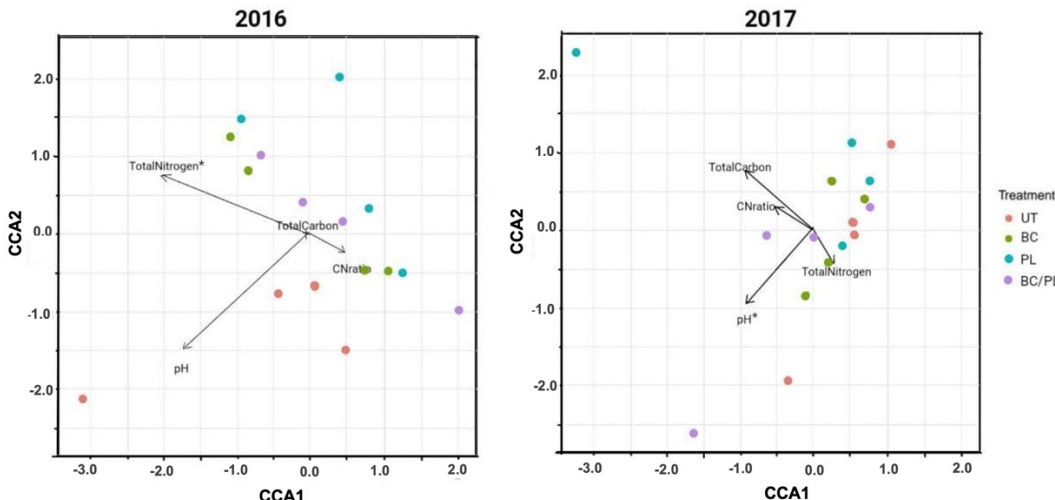

**Figure 2.** Canonical correspondence analysis (CCA) plots between the soil properties and bulk soil community structure of the soybean for 2016 and 2017, showing the vector of continuous environmental variables scaled by their correlation. Longer arrows denote "strong" predictors and the asterisk (*) indicates soil properties significantly correlated with the microbial community structure (BIOENV and Adonis test, $p \leq 0.05$).

In our findings, the application of soil amendments in soybean fields showed 12% to 19% variation in the microbial communities affected by total N and pH in the first and second year, respectively (Figure 2). However, total N was strongly associated with the microbial community composition in the first year ($r_s = 0.74$), while the combination of pH and total C was moderately correlated ($r_s = 0.42$) in the second year.

### 3.4. Bacterial Community Composition and Differential Abundance

Proteobacteria, Acidobacteria, Bacteroidetes, Verrucomicrobia, and Gemmatimonadetes were the five dominant phyla across treatments over the 2 years (Table 8 and Figure 3). Phyla that comprise putative beneficial bacteria, including Nitrospirae, Actinobacteria, Planctomycetes, Chloroflexi, and Firmicutes were also found to be major ones (Figure 3a). There was no statistical difference among treatments at the phylum level in both years, except for Actinobacteria (Kruskal–Wallis test: $\chi^2 = 9.55$, $p = 0.022$) and Chloroflexi (Kruskal–Wallis test: $\chi^2 = 9.22$, $p = 0.026$), which were more abundant in the untreated control in the first year of this study (Table 8 and Figure 3a). Interestingly, however, the phylum Bacteroidetes increased in the second year compared to the first year in all soil samples, including the untreated control (Figure 3a).

**Table 8.** Statistical data of one-way ANOVA (*p*-values) and Kruskal–Wallis test ($\chi^2$ and *p*-values) for the bacterial abundance at the phylum, order, and genus level.

| Taxonomy | ANOVA | | Kruskal–Wallis Test | | | |
| | Year 1 | Year 2 | Year 1 | | Year 2 | |
| | *p* | *p* | $\chi^2$ | *p* | $\chi^2$ | *p* |
|---|---|---|---|---|---|---|
| Phylum | | | | | | |
| Proteobacteria | 0.113 | 0.188 | 6.15 | 0.104 | 4.85 | 0.183 |
| Acidobacteria | 0.781 | 0.317 | 0.55 | 0.907 | 2.47 | 0.481 |
| Bacteroidetes | 0.167 | 0.074 | 4.3 | 0.23 | 5.69 | 0.128 |
| Verrucomicrobia | 0.571 | 0.438 | 2.8 | 0.423 | 3.53 | 0.317 |
| Gemmatimonadetes | 0.424 | 0.127 | 3.07 | 0.382 | 5.89 | 0.117 |
| Thaumarchaeota | 0.571 | 0.410 | 2.27 | 0.518 | 1.28 | 0.734 |
| Nitrospira | 0.960 | 0.252 | 0.19 | 0.978 | 3.24 | 0.356 |
| Actinobacteria | 0.0006 | 0.277 | 9.55 | 0.022 | 4.96 | 0.175 |
| Chloroflexi | 0.0014 | 0.211 | 9.22 | 0.026 | 4.76 | 0.189 |
| Planctomycetes | 0.308 | 0.522 | 2.86 | 0.415 | 2.67 | 0.446 |
| Firmicutes | 0.088 | 0.399 | 7.35 | 0.062 | 4.08 | 0.253 |
| Cyanobacteria | 0.625 | 0.645 | 1.57 | 0.667 | 1.83 | 0.608 |
| Class | | | | | | |
| Sphingobacteriia | 0.375 | 0.078 | 4.48 | 0.214 | 4.65 | 0.199 |
| Blastocatellia | 0.578 | 0.146 | 1.88 | 0.599 | 5.18 | 0.159 |
| Alphaproteobacteria | 0.048 | 0.239 | 7.13 | 0.680 | 3.62 | 0.306 |
| Betaproteobacteria | 0.258 | 0.441 | 2.96 | 0.399 | 1.48 | 0.687 |
| Deltaproteobacteria | 0.439 | 0.312 | 2.89 | 0.409 | 3.15 | 0.368 |
| Gammaproteobacteria | 0.378 | 0.039 | 3.99 | 0.262 | 7.88 | 0.049 |
| Acidobacteria_Subgroup 6 | 0.450 | 0.226 | 2.34 | 0.505 | 4.04 | 0.258 |
| Soil Crenarchaeotic Group (SCG) | 0.159 | 0.407 | 4.13 | 0.248 | 1.70 | 0.637 |
| Gemmatimonadetes | 0.522 | 0.323 | 3.60 | 0.309 | 4.12 | 0.248 |
| Nitrospira | 0.960 | 0.252 | 0.20 | 0.978 | 3.24 | 0.356 |
| OPB35 soil group | 0.440 | 0.857 | 3.20 | 0.362 | 1.48 | 0.687 |
| Spartobacteria | 0.915 | 0.248 | 0.49 | 0.922 | 5.85 | 0.119 |
| Cytophagia | 0.795 | 0.096 | 0574 | 0.903 | 7.21 | 0.065 |
| Solibacteres | 0.301 | 0.705 | 3.51 | 0.320 | 1.32 | 0.724 |
| Unassigned | 0.029 | 0.624 | 8.36 | 0.039 | 1.52 | 0.677 |
| Order | | | | | | |
| Sphingobacteriales | 0.374 | 0.078 | 4.48 | 0.214 | 4.65 | 0.198 |
| Blastocatellales | 0.578 | 0.146 | 1.87 | 0.599 | 5.18 | 0.158 |
| Sphingomonadales | 0.180 | 0.362 | 3.81 | 0.282 | 3.73 | 0.292 |
| Burkholderiales | 0.065 | 0.62 | 7.23 | 0.064 | 1.30 | 0.729 |
| Xanthomonadales | 0.148 | 0.008 | 5.58 | 0.133 | 9.33 | 0.025 |
| Gemmatimonadales | 0.522 | 0.329 | 3.60 | 0.308 | 4.12 | 0.248 |
| Nitrospirales | 0.959 | 0.252 | 0.20 | 0.977 | 3.24 | 0.355 |
| Rhizobiales | 0.011 | 0.226 | 7.56 | 0.056 | 5.85 | 0.119 |
| Nitrosomonadales | 0.94 | 0.631 | 0.29 | 0.962 | 2.18 | 0.535 |
| Myxococcales | 0.01 | 0.033 | 8.58 | 0.035 | 8.14 | 0.043 |
| Verrucomicrobia_OPB35uncultured | 0.515 | 0.866 | 3.24 | 0.356 | 1.88 | 0.599 |
| SCG_uncultured bacterium | 0.027 | 0.745 | 8.49 | 0.037 | 0.82 | 0.846 |
| Chthoniobacterales | 0.915 | 0.248 | 0.48 | 0.922 | 5.85 | 0.119 |
| Cytophagales | 0.794 | 0.095 | 0.57 | 0.902 | 7.21 | 0.065 |
| Desulfurellales | 0.889 | 0.465 | 1.12 | 0.771 | 2.38 | 0.497 |
| Solibacterales | 0.3 | 0.704 | 3.50 | 0.320 | 1.32 | 0.724 |
| Subgroup6_uncultured | 0.429 | 0.2 | 1.61 | 0.657 | 4.32 | 0.228 |
| Subgroup6_ambiguous_taxa | 0.583 | 0.196 | 1.70 | 0.637 | 4.79 | 0.188 |
| Rhodospirillales | 0.421 | 0.605 | 2.36 | 0.501 | 1.30 | 0.729 |
| Unassigned | 0.028 | 0.623 | 8.36 | 0.039 | 1.52 | 0.677 |
| Family | | | | | | |

**Table 8.** *Cont.*

| Taxonomy | ANOVA | | Kruskal–Wallis Test | | | |
| | Year 1 | Year 2 | Year 1 | | Year 2 | |
| | $p$ | $p$ | $\chi^2$ | $p$ | $\chi^2$ | $p$ |
| Blastocatellaceae (subgroup 4) | 0.578 | 0.146 | 1.88 | 0.599 | 5.18 | 0.159 |
| Chitinophagaceae | 0.297 | 0.037 | 3.64 | 0.303 | 6.51 | 0.089 |
| Sphingomonadaceae | 0.105 | 0.369 | 4.35 | 0.227 | 2.92 | 0.402 |
| Sphingobacteriales_env.OPS | 0.919 | 0.241 | 0.51 | 0.917 | 4.36 | 0.225 |
| Gemmatimonadaceae | 0.522 | 0.329 | 3.60 | 0.309 | 4.13 | 0.248 |
| Nitrospiraceae | 0.860 | 0.273 | 0.11 | 0.991 | 3.24 | 0.356 |
| Nitrosomonadaceae | 0.935 | 0.621 | 0.33 | 0.954 | 2.18 | 0.535 |
| Verrrucomicrobia_OPB35 group | 0.516 | 0.866 | 3.24 | 0.356 | 1.88 | 0.599 |
| Xanthomonadales Incertae Sedis | 0.036 | 0.021 | 7.79 | 0.051 | 7.17 | 0.067 |
| Comamonadaceae | 0.060 | 0.455 | 7.21 | 0.065 | 2.98 | 0.395 |
| Soil Crenarchaetotic Group (SCG) | 0.623 | 0.745 | 1.83 | 0.608 | 0.82 | 0.846 |
| Oxalobacteraceae | 0.153 | 0.714 | 6.37 | 0.087 | 0.81 | 0.845 |
| Cytophagaceae | 0.806 | 0.104 | 6.57 | 0.087 | 6.55 | 0.088 |
| Desulfurellaceae | 0.889 | 0.465 | 1.13 | 0.771 | 2.38 | 0.497 |
| Solibacteraceae (subgroup 3) | 0.301 | 0.705 | 3.51 | 0.320 | 1.33 | 0.724 |
| Subgroup 6_uncultured | 0.429 | 0.196 | 1.61 | 0.657 | 4.79 | 0.188 |
| Subgroup6_Ambiguous_taxa | 0.672 | 0.227 | 2.05 | 0.562 | 2.14 | 0.544 |
| Unassigned | 0.029 | 0.624 | 8.36 | 0.030 | 1.52 | 0.677 |

The most abundant classes were Sphingobacteria in the phylum Bacteroidetes; Blastocatellia and subgroup 6 in the phylum Acidobacteria; and classes in the phylum Proteobacteria including Alphaproteobacteria, Betaproteobacteria, Deltaproteobacteria, and Gammaproteobacteria (Table 8 and Figure 3b). Other major classes were Gemmatimonadetes, Nitrospira, and the Soil Crenarchaeotic Group (SCG). There was no significant difference observed among the classes in the first year. However, in the second year, Gammaproteobacteria was more abundant in the BC/PL-treated soil than the untreated soil ($p = 0.049$) (Figure 3b).

The top 20 orders with a relative abundance of >1% were analyzed using the Kruskal–Wallis test (Table 8 and Figure 3c). In the first year, Myxococcales and SCG uncultured bacterium were both abundant in the untreated and BC-treated soil ($p < 0.05$) (Figure 3c). During the second year, Xanthomonadales ($p = 0.008$) and Myxococcales ($p = 0.043$) were significantly more abundant in the BC/PL- and PL-treated soils (Figure 3c). At the family level, unassigned taxa ($p = 0.030$) were significantly higher in the untreated soil in the first year, while the *Chitinophagaceae* ($p = 0.037$) was significantly higher in the BC-treated soil in the second year (Figure 3d). There were significant differences in the microbial composition from 2016 to 2017, which may be attributed to the proliferation of bacterial taxa favored by the addition of soil amendments, and thus a shift in the microbial community over time.

At the genus level, we identified features strongly influenced by different soil treatments, which were enriched or depleted genera based on relative abundance as compared to the untreated soil. There were 166, 214, and 178 enriched genera (primarily the identifiable genera from the phyla Proteobacteria, Bacteroidetes, Actinobacteria, Acidobacteria, and Planctomycetes (Table 9)), and 154, 234, and 197 depleted genera (primarily from the phyla Proteobacteria, Actinobacteria, Bacteroidetes, Chloroflexi and Firmicutes (Table 10)) in the BC-, PL-, and BC/PL-treated soils, respectively (Figure 4).

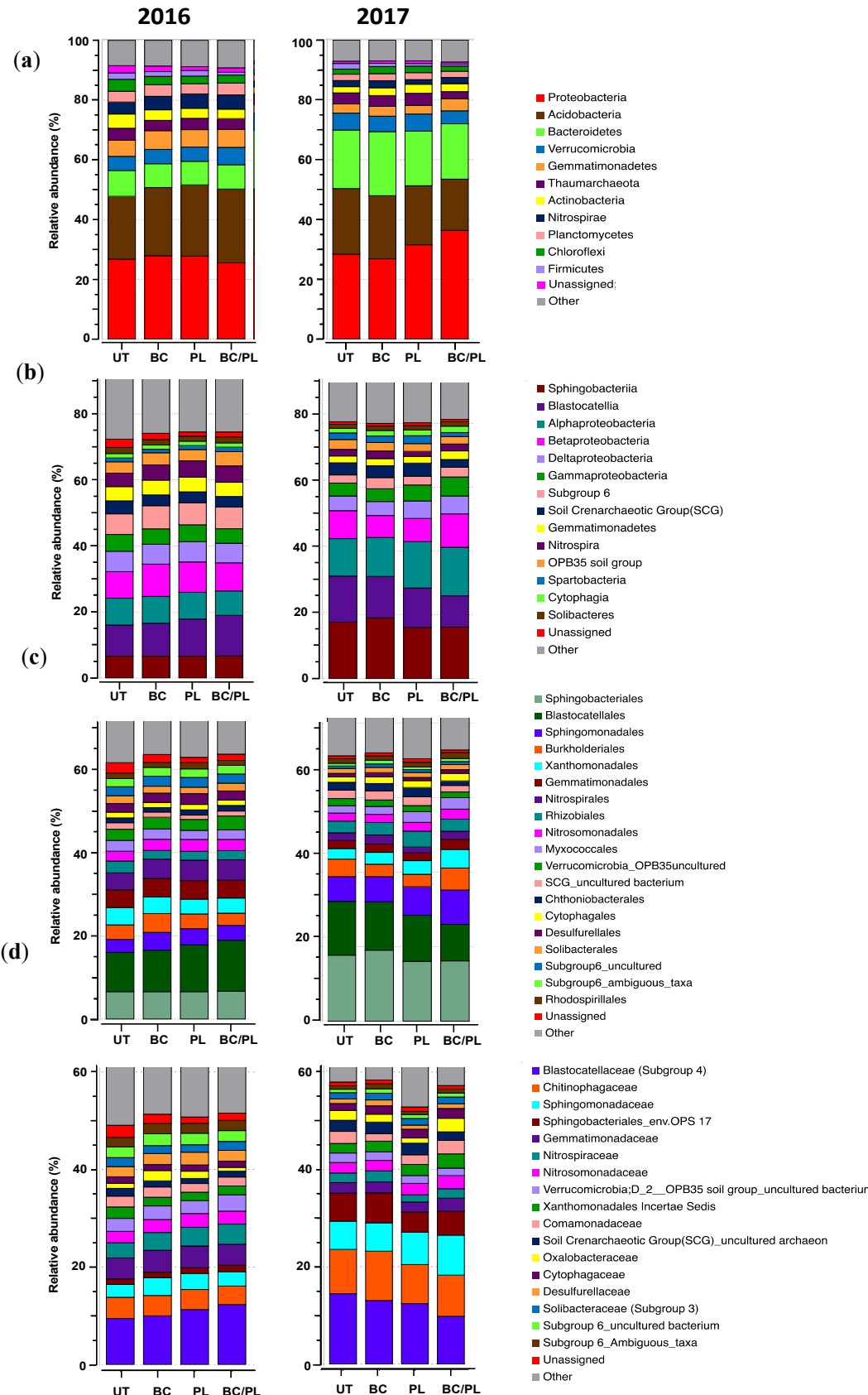

**Figure 3.** Bacterial taxonomic composition of the treated and untreated soils at the: (**a**) phylum, (**b**) class, (**c**) order, and (**d**) family levels as influenced by different soil treatments over the 2 growing seasons.

**Table 9.** The ASVs at the genus level enriched by different soil treatments.

| | | | 2016 | | |
|---|---|---|---|---|---|
| | **Phylum** | **Count** | **Enriched ASVs \*** | **Log2 FC** | ***p*-Adjusted** |
| | Proteobacteria | 17 | *Nitrospirae_uncultured* | 4.487 | 0.009 |
| | Bacteroidetes | 9 | *Acinetobacter* | 5.711 | 0.022 |
| | Actinobacteria | 5 | *Rivibacter* | 5.839 | 0.027 |
| | Verrucomicrobia | 3 | *Chlamydiales* | 4.322 | 0.054 |
| | Planctomycetes | 3 | *Verrucomicrobia_uncultured* | 4.823 | 0.060 |
| BC vs. UT | Gemmatimonadetes | 2 | *Spirochaetae_Turneriella* | 4.810 | 0.060 |
| | Nitrospirae | 2 | *Uncultured_Pseudolabrys* | 4.221 | 0.062 |
| | Acidobacteria | 2 | *Crocinitomix* | 4.143 | 0.062 |
| | Others | 10 | *Latescibacteria_uncultured* | 3.436 | 0.065 |
| | Total | 53 | *Uncultured_alphaproteobacterium* | 4.760 | 0.062 |
| | Proteobacteria | 28 | *Chthoniobacterales_DA101* | 5.349 | 0.016 |
| | Acidobacteria | 10 | *Sphingobacterium* | 5.409 | 0.029 |
| | Bacteroidetes | 6 | *Uncultured euyarchaeote* | 6.523 | 0.037 |
| | Planctomycetes | 4 | *Acidobacteria_uncultured* | 5.093 | 0.037 |
| | Verrucomicrobia | 4 | *Acinetobacter* | 4.467 | 0.037 |
| PL vs. UT | Thaumarcheota | 3 | *Uncultured alpha proteobacterium* | 4.444 | 0.037 |
| | Nitrospirae | 2 | *Rhodocista* | 4.016 | 0.046 |
| | Firmicutes | 2 | *Uncultured_Pseudolabrys* | 4.293 | 0.049 |
| | Others | 10 | *Proteobacteria_Delftia* | 4.598 | 0.052 |
| | Total | 69 | *Rhodospirillales_uncultured* | 3.987 | 0.052 |
| | Proteobacteria | 16 | *Chthoniobacterales_DA101* | 5.692 | 0.004 |
| | Planctomycetes | 5 | *Flavobacteriales_uncultued* | 4.780 | 0.022 |
| | Acidobacteria | 5 | *Sphingobacterium* | 4.170 | 0.022 |
| | Bacteroidetes | 4 | *Duganella* | 4.644 | 0.030 |
| BC/PL vs. UT | Verrucomicrobia | 4 | *Planctomycetes* | 4.074 | 0.042 |
| | Nitrospirae | 2 | *Prosthecobacter* | 4.179 | 0.050 |
| | Actinobacteria | 2 | *Acidobacteria_uncultured* | 4.010 | 0.062 |
| | Gemmatimonadetes | 2 | *Proteobacteria_Leptothrix* | 4.358 | 0.067 |
| | Others | 10 | *Proteobacteria_uncultured* | 4.005 | 0.067 |
| | Total | 50 | *Myxococcales_uncultured* | 4.005 | 0.067 |
| | | | 2017 | | |
| | **Phylum** | **Count** | **Enriched ASVs \*** | **Log2 FC** | ***p*-Adjusted** |
| | Proteobacteria | 46 | *Pelomonas* | 6.563 | 3.06E-05 |
| | Cyanobacteria | 16 | *Paenarthrobacter* | 4.662 | 0.007 |
| | Bacteroidetes | 13 | *Proteobacteria_uncultured* | 5.725 | 0.009 |
| | Actinobacteria | 10 | *Proteobacteria_uncultured* | 5.054 | 0.016 |
| BC vs. UT | Acidobacteria | 7 | *Bacteroidetes_Emticicia* | 4.857 | 0.025 |
| | Firmicutes | 4 | *Acidobacteria_uncultured* | 5.143 | 0.035 |
| | Planctomycetes | 3 | *Cyanobacteria_uncultured* | 4.964 | 0.045 |
| | Nitrospirae | 2 | *Ohtaekwangia* | 1.461 | 0.045 |
| | Others | 12 | *Proteobacteria_uncultured* | 4.419 | 0.047 |
| | Total | 113 | *Leptospirillum* | 4.534 | 0.068 |
| | Proteobacteria | 76 | *Nannocystis* | 3.494 | 0.001 |
| | Actinobacteria | 17 | *Cellulosimicrobium* | 6.237 | 0.001 |
| | Bacteroidetes | 13 | *Paenarthrobacter* | 4.902 | 0.001 |
| | Cyanobacteria | 13 | *Proteobacteria_uncultured* | 7.134 | 0.006 |
| PL vs. UT | Firmicutes | 6 | *Luteimonas* | 5.551 | 0.007 |
| | Acidobacteria | 4 | *Stigmatella* | 5.840 | 0.007 |
| | Gemmatimonadetes | 2 | *Lysinimonas* | 5.209 | 0.007 |
| | Armatimonadetes | 2 | *Proteobacteria_Devosia* | 2.187 | 0.009 |
| | Others | 12 | *Chitinimonas* | 5.587 | 0.010 |
| | Total | 145 | *Actinobacteria_Pilimelia* | 5.761 | 0.016 |

**Table 9.** *Cont.*

| | Phylum | Count | Enriched ASVs * | Log2 FC | *p*-Adjusted |
|---|---|---|---|---|---|
| | **2016** | | | | |
| | Proteobacteria | 75 | *Chitinimonas* | 7.058 | 0.025 |
| | Actinobacteria | 14 | *Proteobacteria_uncultured* | 5.718 | 0.025 |
| | Bacteroidetes | 10 | *Proteobacteria_Minicystis* | 5.549 | 0.025 |
| | Cyanobacteria | 7 | *Bacteroidetes_Emticicia* | 5.404 | 0.033 |
| BC/PL vs. UT | Gemmatimonadetes | 5 | *Proteobacteria_uncultured* | 5.770 | 0.033 |
| BC vs. UT | Firmicutes | 4 | *Noviherbaspirillum* | 5.288 | 0.033 |
| | Planctomycetes | 4 | *Actinobacteria_Lentzea* | 5.170 | 0.035 |
| | Acidobacteria | 3 | *Comamonas* | 1.892 | 0.035 |
| | Others | 6 | *Proteobacteria_Pelomonas* | 5.468 | 0.049 |
| | Total | 128 | *Proteobacteria_uncultured* | 4.873 | 0.053 |

* The 10 most enriched ASVs for each treatment. The detailed taxonomic index of each ASV is presented in Supplementary Table S2.

**Table 10.** The ASVs at the genus level depleted by different soil treatments.

| | Phylum | Count | Depleted ASVs * | Log2 FC | *p*-Adjusted |
|---|---|---|---|---|---|
| | **2016** | | | | |
| | Proteobacteria | 34 | *Thaumarchaeota_uncultured* | −6.554 | 0.001 |
| | Actinobacteria | 12 | *Cellulomonas* | −4.917 | 0.001 |
| | Bacteroidetes | 5 | *Erwinia* | −7.011 | 0.010 |
| | Chloroflexi | 5 | *Cyanobacteria* | −4.809 | 0.018 |
| BC vs. UT | Firmicutes | 4 | *Proteobacteria_Zymoseptoria* | −4.165 | 0.027 |
| | Cyanobacteria | 3 | *Proteobacteria_Lecanicillium* | −4.500 | 0.027 |
| | Elusimicrobia | 3 | *Proteobacteria_uncultured* | −3.871 | 0.027 |
| | Armatimonadetes | 2 | *Firmicutes_Clostridium* | −4.648 | 0.033 |
| | Others | 11 | *Flavobacterium* | −4.679 | 0.037 |
| | Total | 79 | *Chloroflexi* | −4.155 | 0.037 |
| | Proteobacteria | 62 | *Proteobacteria_Metarhizium* | −4.314 | <0.000 |
| | Actinobacteria | 21 | *Thaumarchaeota_uncultured* | −6.476 | <0.000 |
| | Chloroflexi | 17 | *Bradyrhizobium* | −2.117 | 0.002 |
| | Bacteroidetes | 9 | *Cellulomonas* | −4.852 | 0.002 |
| PL vs. UT | Acidobacteria | 8 | *Hypsibius* | −6.149 | 0.002 |
| | Cyanobacteria | 8 | *Proteobacteria uncultured* | −2.946 | 0.015 |
| | Planctomycetes | 5 | *Roseiflexus* | −1.459 | 0.016 |
| | Parcubacteria | 5 | *Dactylosporangium* | −1.767 | 0.017 |
| | Others | 21 | *Caldithrix* | −2.679 | 0.029 |
| | Total | 156 | *Cyanobacteria* | −3.207 | 0.029 |
| | Proteobacteria | 58 | *Proteobacteria_Zymoseptoria* | −3.803 | <0.000 |
| | Actinobacteria | 22 | *Caldithrix* | −5.662 | <0.000 |
| | Chloroflexi | 12 | *Actinobacteria_Asanoa* | −5.777 | <0.000 |
| | Firmicutes | 9 | *Thaumarchaeota_uncultured* | −6.496 | <0.000 |
| | Bacteroidetes | 4 | *Chloroflexi_Roseiflexus* | −1.432 | <0.000 |
| BC/PL vs. UT | Cyanobacteria | 3 | *Firmicutes_Tumebacillus* | −2.191 | <0.000 |
| | Parcubacteria | 3 | *Nitrospirae_uncultured* | −3.682 | <0.000 |
| | Thaumarcheota | 3 | *Proteobacteria_uncultured* | −6.088 | <0.000 |
| | Others | 20 | *Bradyrhizobium* | −2.112 | <0.000 |
| | Total | 134 | *Virgisporangium* | −5.750 | <0.000 |

**Table 10.** *Cont.*

| | Phylum | Count | Depleted ASVs * | Log2 FC | *p*-Adjusted |
|---|---|---|---|---|---|
| | **2017** | | | | |
| BC vs. UT | Proteobacteria | 35 | *Pseudogulbenkiania* | −7.155 | 0.000 |
| | Firmicutes | 13 | *Dechloromonas* | −7.262 | 0.000 |
| | Actinobacteria | 7 | *Paucimonas* | −6.845 | 0.000 |
| | Bacteroidetes | 3 | *Aquincola* | −6.457 | 0.000 |
| | Acidobacteria | 3 | *Dechlorobacter* | −6.209 | 0.000 |
| | Cyanobacteria | 2 | *Proteobacteria_uncultured* | −5.666 | 0.001 |
| | Elusimicrobia | 2 | *Bacteroidetes_uncultured* | −1.382 | 0.008 |
| | Siprochaetes | 2 | *Firmicutes_Clostridium* | −5.981 | 0.022 |
| | Others | 8 | *Azotobacter* | −7.282 | 0.022 |
| | Total | 75 | *Lachnoclostridium* | −6.377 | 0.045 |
| PL vs. UT | Proteobacteria | 27 | *Firmicutes_Clostridium* | −6.232 | 0.000 |
| | Firmicutes | 18 | *Herbaspirillum* | −7.281 | 0.004 |
| | Bacteroidetes | 11 | *Geobacter* | −2.536 | 0.006 |
| | Actinobacteria | 4 | *Gemmatimonadetes_AKAU4049* | −5.408 | 0.006 |
| | Gemmatimonadetes | 3 | *Caenimonas* | −5.876 | 0.006 |
| | Cyanobacteria | 3 | *Bacteroidetes_Pedobacter* | −5.986 | 0.006 |
| | Acidobacteria | 2 | *Firmicutes_Clostridium* | −6.988 | 0.007 |
| | Chloroflexi | 2 | *Proteobacteria_uncultured* | −5.171 | 0.008 |
| | Others | 8 | *Gemmatimonadetes_bacterium* | −6.134 | 0.008 |
| | Total | 78 | *Sporacetigenium* | −5.354 | 0.010 |
| BC/PL vs. UT | Firmicutes | 17 | *Bacteroidetes_Pedobacter* | −5.868 | 0.025 |
| | Proteobacteria | 16 | *Proteobacteria_uncultured* | −5.075 | 0.027 |
| | Actinobacteria | 6 | *Gemmatimonadetes_bacterium* | −5.984 | 0.027 |
| | Bacteroidetes | 5 | *Azotobacter* | −7.199 | 0.033 |
| | Cyanobacteria | 4 | *Cyanobacteria_Calothrix* | −5.306 | 0.053 |
| | Verrucomicrobia | 4 | *Sedimentibacter* | −6.200 | 0.056 |
| | Acidobacteria | 3 | *Lachnoclostridium* | −6.293 | 0.056 |
| | Chlamydia | 2 | *Clostridium* | −5.087 | 0.060 |
| | Others | 6 | *Elusimicrobia_uncultured* | −3.052 | 0.079 |
| | Total | 63 | *Verrucomicrobia_uncultured* | −3.077 | 0.095 |

* The 10 most depleted ASVs by each treatment. The detailed taxonomic index of each ASV is presented in Supplementary Table S3.

In the first year, the most significant enriched genera in the BC-treated soils were *Nitrospira*_uncultured, *Acinetobacter*, and *Rivibacter* (Table 9), while depleted genera were *Cellumonas*, *Erwinia*, *Clostridium*, *Flavobacterium*, and uncultured members of Proteobacteria, Chloroflexi, and Thaumarcheota (Table 10). The PL-treated soil was also enriched with *Acinetobacter*, *Sphingobacterium*, *Rhodocista*, and other uncultured members of the phyla, as well as Proteobacteria, Verrucomicrobia, and Thaumarcheota (Table 9). Meanwhile, *Bradyrhizobium, Roseiflexus, Dactylosporangium, Caldithrix,* and *Cyanobacteria* were depleted in the PL-treated soil (Table 10). The enriched genera in BC/PL-treated soil included the *Chthoniobacterales_DA101* soil group, *Flavobacteriales_uncultured*, *Sphingobacterium, Duganella, Prosthecobacter*, and other uncultured members of Verrucomicrobia and Bacteroidetes, while *Caldithrix, Asanoa, Roseiflexus, Tumebacillus, Bradyrhizobium, Virgisporangium*, and uncultured members of Proteobacteria and Nitrospirae were identified as depleted genera (Tables 9 and 10). There were much fewer enriched and depleted genera in the BC-treatment soil as compared to the PL- and BC/PL-treated soils. In the second year, the most significant enriched genera in the treated soils were *Pelomonas, Paenarthtobacter, Emticicia, Chitinimonas, Nannocystis, Cellulosimicrobium, Luteimonas, Stigmatella, Lysinimonas*, and several members of Proteobacteria (Table 9), while the main depleted genera were *Clostridium, Pedobacter, Nitrospira, Sphingomonas, Pseudogulbenkiania, Dechloromonas, Paucimonas, Aquincola, Dechlorobacter*, and several members of Proteobacteria and Gemmatimonadetes (Table 10). Notably, enriched genera in the treated soils increased in the second year regardless of the treatment, in which the PL-treated soil contained the highest number

of enriched genera (Table 9 and Figure 4), suggesting that the 2 years of treatment caused the enrichment of certain bacterial groups in the treated soil.

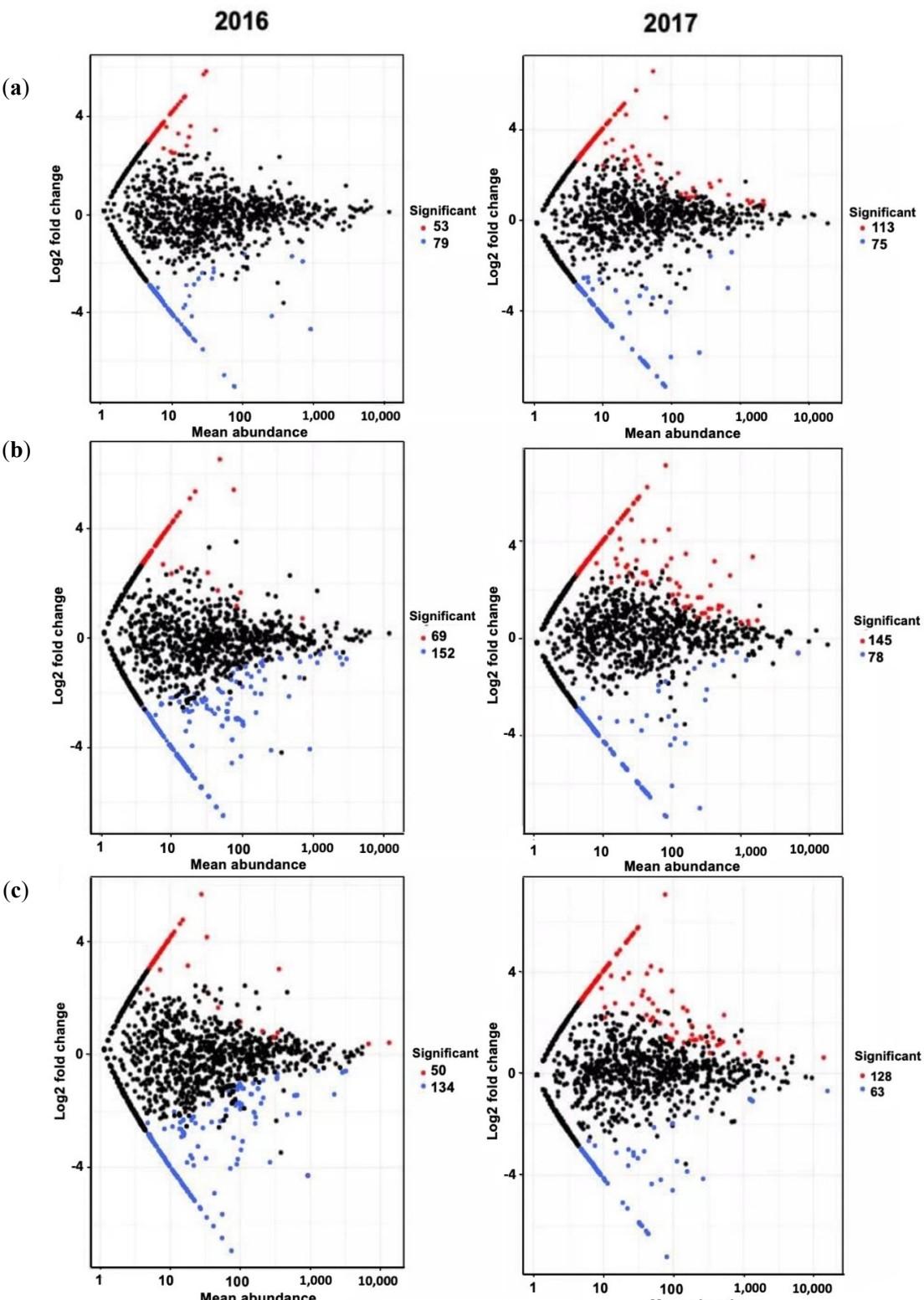

**Figure 4.** Mean abundance plots illustrating ASVs significantly enriched (red) and depleted (blue) by soil treatment. Volcano plots compared with untreated control as determined by differential abundance analysis. Each point represents the individual ASVs, and the Y axis indicates the abundance fold change vs. control. (**a**) BC-treated soil vs. control; (**b**) PL-treated soil vs. control; (**c**) BC/PL-treated soil vs. control in 2016 and 2017, respectively.

### 3.5. Microbial Network Analysis

The co-occurrence network of the microbial community in each treatment and year was constructed to assess the impact of soil treatments on the structure of the microbial community (Figure 5 and Figure S4). The entire networks across all treatments in the 2 years were composed of 15,328 significant associations (edges) with 373 nodes, with an average clustering coefficient of 0.68 and an average path length of 2.81 (detailed data not shown). Network edges were comprised mainly of strong positive associations. The key taxa that regulate the network structure were identified based on the node degree (>50), closeness centrality (>0.44), and betweenness centrality scores (<0.18) [64]. These taxa include Verrucomicrobia, Gemmatimonas, Bacteroidetes (env_OPS1_uncultured bacterium), and Acidobacteria (Blastocatellaceae_11-24 and subgroup 4).

In the first year, based on top 35 most abundant taxa, the PL-treated and BC-treated soils had the highest co-occurring taxa belonging to Proteobacteria (including *Nitrosomonadaceae*_uncultured bacterium, *Sphingomonas*, *Desulfurellaceae_H16*, *Rhodospirillales*_uncultured, *Steroidobacter*, and *Ramlibacter*). Meanwhile, the untreated control and BC/PL-treated soils were predominantly composed of co-occurring taxa belonging to Acidobacteria (including uncultured Acidobacteria bacterium, *Candidatus Solibacter*, and *Blastocatellaceae* uncultured bacteria). However, the BC/PL-treated soil had the greatest number of co-occurring taxa under the phylum Gemmatimonadetes (*Gemmatimonas*, *Longimicrobia*, and uncultured Planctomycetes). In the second year, there was an increase in co-occurring taxa in the phylum Bacteroidetes across treatments, particularly in the BC-treated soil (*Sphingobacteria, Chitinophaga, Cytophaga*, and other uncultured Bacteroidetes). The BC/PL-treated soil had the greatest number of co-occurring taxa belonging to phylum Proteobacteria (*Sphingomonas, Nitrosomonadaceae* uncultured, *Steroidobacter, Rhodopirillales, Masillia,* and *Desulfurellales*). The PL-treated soil had the greatest number of co-occurring taxa within Verrucomicrobia (Verrucomicrobia uncultured, *Spartobacteria, Chtoniobacter, and Opitutus*) (Figure 5). The co-occurring taxa within Gemmatimonadetes, Bacteroidetes, Verrucomicrobia, and some members in the Acidobacteria are involved in degrading complex materials. At the same time, co-occurring taxa within Proteobacteria are involved in nutrient cycling in the soil.

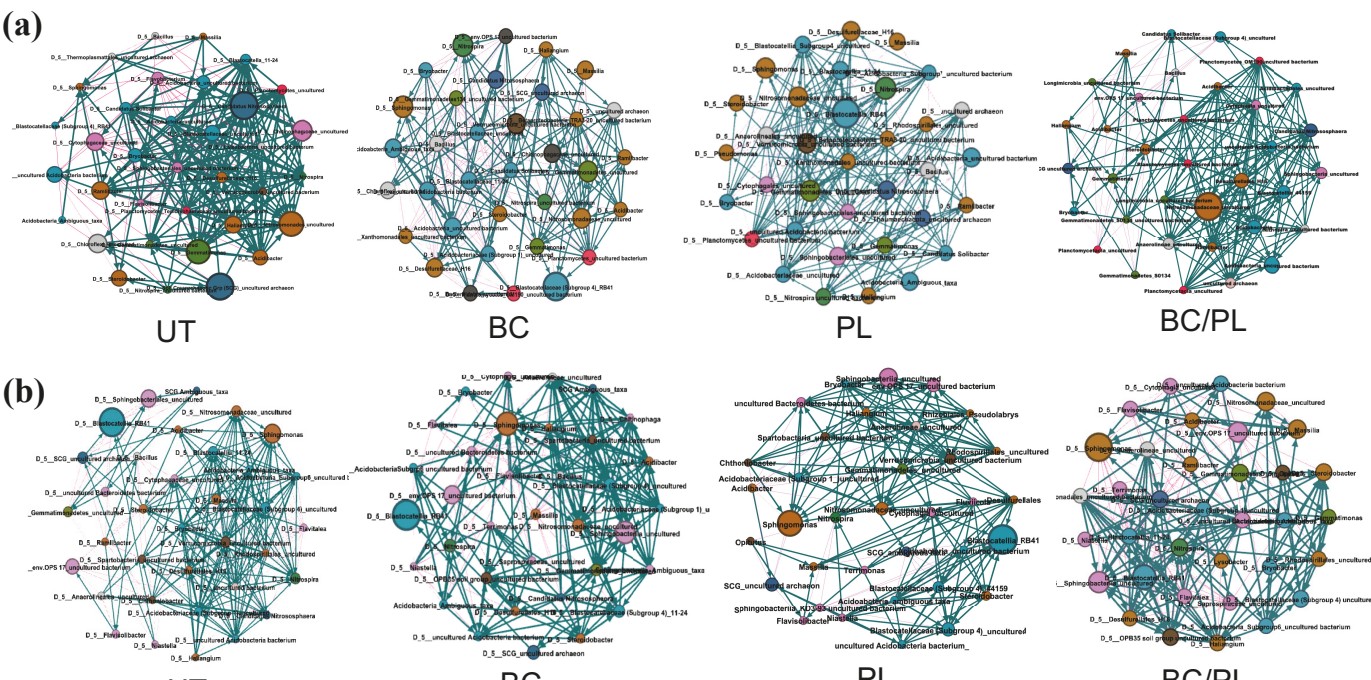

**Figure 5.** *Cont.*

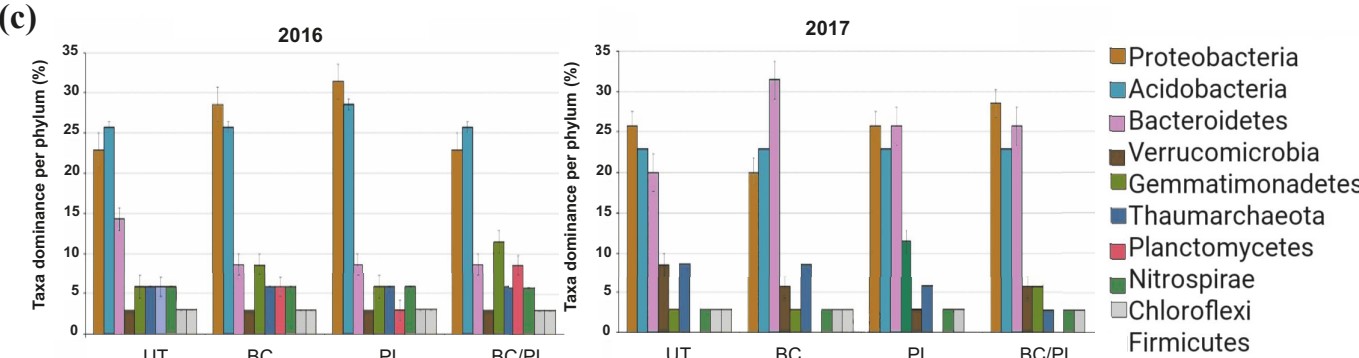

**Figure 5.** Microbial interaction networks of dominant taxa at the genus level (top 35) in the bulk soil in (**a**) 2016 and (**b**) 2017 for untreated (UT) soil and soils treated with biochar (BC), poultry litter (PL), and a combination of biochar and poultry litter (BC/PL). The size of the nodes shows taxa abundance, and the different colors indicate the corresponding taxonomic assignment at the phylum level. The edge color represents positive (green) and negative (pink) correlations. The edge thickness indicates the correlation values; only significant interactions are shown (r > 0.6; *p* < 0.01). (**c**) Taxa dominance (Top 10) in the microbial network per phylum.

## 4. Discussion

Several studies associate soil health and crop productivity with below-ground bacterial diversity [5,15,24–27,30,65]. Following this line of thought, we hypothesized changes in microbial communities due to cultivation management, specifically the application of different soil amendments in the soybean field. The study revealed the gradual impact of soil treatments with BC, PL, and the BC/PL on the bacterial communities and the agronomic traits of soybean plants in terms of growth and yield. Specifically, the growth was significantly highest in the PL-treated soil in the second year and was comparable to that of the BC/PL-treated soil. Biochar characteristics depend upon what it is made from and how it was processed. Most biochar has a small labile fraction of carbon but a much larger recalcitrant fraction that can persist in soils [66]. The biochar used in this study was pinewood biochar (similar to that of a study on *Pinus radiata* biochar), and tends to be recalcitrant and mineralizes in soil slowly [67]. Treatment with BC alone was comparable to no treatment, despite the higher C content it provided, indicating that the addition of biochar alone did not add appreciable amounts of nutrients to the soil. The PL had high total N, P, K, micronutrients, and organic matter (Table S1). Similar to previous studies, the PL-treated soil showed less crop yield in the first and second years of application [17]. Of note, the BC/PL treatment combination showed the highest crop yield for the 2 years, although it was not statistically different from the other treatments.

Alpha diversity indicates no significant difference within treatments. Nevertheless, the species richness and diversity slightly shifted over the 2 years among treatments, suggesting that the microbiomes in the field change over time. Other research groups have shown that biochar causes a slight shift in bacterial community compositions but has no significant short-term impact on microbial diversity in soil environments including maize fields and pasture soils [67–70]. It is still unknown whether the long-term application of biochar can increase microbial diversity [71–73]. However, biochar treatment was also reported to significantly alter soybean field microbial abundance and community composition at a low pH soil condition in one growing season. This was more obvious in the soybean rhizosphere than the bulk soil [74]. Similarly, in the acidic paddy soil, it was reported that the microbial community in the rice rhizosphere responded to biochar treatment more sensitively than bulk soil [13]. These studies suggest that the microbial community in the bulk soil compartment is likely affected gradually by biochar treatment, while it has more immediate effects on the microbial community in the rhizosphere. In our study, soil pH tended to be neutral in both years regardless of treatments (Table 6), although the untreated and poultry litter decreased in pH in the second year.

Regarding the previous studies mentioned above, the neutral soil pH condition could be a reason for the minor variations within the treatments observed in this study. Interestingly, bacterial community structure variation was observed based on phylogenetic-based distance using weighted Unifrac distance, which accounts for the relative abundance of taxa among treatments. Hence, some variations in the lineage of bacterial community composition could be attributed to other factors influenced by soil treatments aside from pH.

pH was found to be an important factor in the community variation of microbial taxa in different agro-ecosystems reported in previous studies [10,75–77]. The availability of N and C was also a main driving factor in the shift of microbial community by different treatments of organic materials [78–80]. Biochar was reported to have a lesser effect on the microbial community structure in neutral or alkaline soil [77]. In contrast to other studies, distinct microbial structures were observed between the biochar-treated and untreated acidic paddy soils [81–83]. In our study, the BC and BC/PL-treated soil maintained a neutral pH, while the PL and untreated soil had a decreased pH in the second year.

With regard to the bacterial community composition, the dominant phyla across treatments were Proteobacteria, Acidobacteria, Bacteroidetes, Verrucomicrobia, and Gemmatimonadetes, consistent with the previous studies on the most dominant phyla in agricultural soils [65,84–86]. These phyla account for 71–81% of total ASVs, implying that these phyla may play an important role in the soybean bulk soil. Cellulolytic bacteria involved in decomposition under orders Myxococcales and Xanthomonadales had a significantly greater presence in the BC/PL- and PL-treated soils in the second year. Members of Myxococcales are considered micropredators in the soil and play a key role in soil C cycling [79]. Interestingly, the differential abundance at the genus level based on adjusted p-values and log-fold change revealed a significant variation between the BC-, PL-, and BC/PL-treated soils vs. untreated soils. The differentially abundant taxa in the treated soils were related to organic matter decomposition and nutrient cycling. Taxa linked to C cycling were *Pelomonas*, *Cellulosimicrobium*, *Chitinomonas Acinetobacter*, *Nannocystis*, and *Stigmatella*. *Pelomonas* species produce enzymes that degrade hemicellulose, carbon substrates, and aromatic compounds [87]. *Chitinomonas* and *Cellulosimicrobium* are chitinolytic bacterium [88,89]. The *Cellulosimicrobium* genome is composed of genes responsible for protection against salinity stresses and production of volatiles, with a number of genes for the degradation of plant biopolymers [90]. *Acinetobacter* have metabolic capabilities for the degradation of various long-chain dicarboxylic acids and aromatic and hydroxylated aromatic compounds that are associated with plant degradation products [91]. *Nannocystis* is a common myxobacteria in soil, and is usually from decaying substrates [92]. *Stigmatella* is another myxobacteria capable of breaking down peptidoglycans, polysaccharides, protein, and other cellular detritus. Moreover, *Stigmatella* produces antibiotics toxic to yeast and filamentous fungi but not most bacteria [93]. On the other hand, differentially abundant taxa in the treated soils associated with N and other nutrient cycling were *Nitrospira_uncultured*, *Duganella, Leptospirillum*, and other indicators of soil fertility such as *Pilimelia, Ramlibacter*, and *Chthoniobacter_DA101* [94]. *Nitrospira* are chemolithoautotrophic organisms capable of complete nitrification [95]. *Duganella* can solubilize phosphorus, potassium, and zinc in soils to promote plant growth [96,97]. Some species of *Leptospirillum*, such as *Leptospirillum ferrooxidans*, are iron-oxidizing bacterium that contain all genes necessary for nitrogen fixation such as Mo-Fe nitrogenase, specific regulator (nifA), global regulators (glnB and ntrC), and other sensors and transport systems related to nitrogen assimilation. On the other hand, most denitrifying bacteria such as *Pseudogulbenkiania, Dechlorobacter*, and *Flavobacterium* were depleted in the treated soils [98–100], while *Rivibacter* [99] was enriched in the BC-treated soil.

The key taxa in the co-occurrence network analysis were members of the Verrucomicrobia, Gemmatimonadetes, Bacteroidetes, and Acidobacteria subgroup 4. Verrucomicrobia members are known as degraders of recalcitrant organic matter [101]. Gemmatimonas is highly abundant in soils with pyrogenic organic matter, which likely decomposes polyaromatic C [82,102]. In addition, these bacteria are capable of decomposing cellulose,

lignocellulose, and chitin in biochar-amended soils [103]. Bacteriodetes are considered as specialists for degrading high molecular weight organic matter and are indicators of nutrient-rich soil [104,105]. Acidobacteria of subgroup 4, especially Blastocatella, were abundant in organic C content [79]. Overall, the co-occurring taxa associated with treated soils and those that regulate the microbiome structure in the treated soils are known to be related to organic matter decomposition and nutrient cycling, suggesting that soils treated with poultry litter and the combination of biochar and poultry litter favor beneficial bacteria for soybean growth.

## 5. Conclusions

This study used high-throughput amplicon sequencing to characterize the soil microbial community profiles in soybean field plots. The community structures were correlated with total N, total C, and pH in the 2 years. The co-occurrence networks revealed key taxa involved in organic matter decomposition and nutrient transformation in the treated soils. The PL treatment had the most profound impact on the microbial community among other treatments, followed by the BC/PL and BC treatments. Our study provides insights into the impact of organic fertilizing materials on the improvement of the microbial community to favor beneficial bacteria that promote plant growth and health. The comparable effect of PL and BC/PL treatments on soybean growth indicates a potential use of BC/PL to prevent nutrient loss and enrich the soil for an ecologically sound crop production strategy. Further studies with extended field trials accompanied by more comprehensive studies of the microbial community structure in the different compartments (e.g., bulk soil, soybean rhizosphere, and root endosphere) would substantially enhance our understanding of the mechanistic basis underlying the beneficial effects of the organic fertilizing materials. Furthermore, a parallel experiment such as a mesocosm study using field soil under controlled conditions would minimize the risk of possible confounding variables in the field.

**Supplementary Materials:** The following are available online at https://www.mdpi.com/article/10.3390/agronomy11071428/s1, Figure S1: Rarefaction plots on the observed ASVs of the different soil treatments at different sampling depths, Figure S2: Box plots of the alpha diversity patterns for the soil samples showing the richness (A) and diversity (B), Figure S3: The beta diversity among soil treatments in 2016 and 2017 depicted in non-metric multidimensional scaling (NMDS) plots with Bray–Curtis dissimilarity distances, unweighted unifrac, and weighted unifrac distance measure among soil treatments: untreated (UT), biochar (BC), poultry litter (PL), and combination of biochar and poultry litter (BC/PL), Figure S4: General co-occurrence networks of the microbial community structure, Table S1: Sequence alignment and relative abundance at different taxonomic levels, Table S2: The detailed taxonomic index of enriched ASVs at the genus level in the different soil treatments, Table S3: The detailed taxonomic index of depleted ASVs in the different soil treatments.

**Author Contributions:** Conceptualization, R.B.C. and J.H.H.; methodology, R.B.C., C.J. and J.H.H.; software, R.B.C., L.M.C. and N.K.; formal analysis, R.B.C., Y.J.J. and L.M.C.; investigation, R.B.C., C.J., H.-H.K. and J.H.H.; resources, J.H.H. and C.J.; data curation, R.B.C.; writing—original draft preparation, R.B.C.; writing—review and editing, J.H.H. and C.J.; visualization, R.B.C. and L.M.C.; supervision, J.H.H.; project administration, J.H.H.; funding acquisition, J.H.H. and C.J. All authors have read and agreed to the published version of the manuscript.

**Funding:** This research was funded by the LSU Economic Development Assistantship, the Fulbright Fellowship, the Louisiana Soybean & Grain Research and Promotion Board (GR-00007089-AWD-003027) and USDA NIFA (Hatch project: LAB94386).

**Data Availability Statement:** The sequence data analyzed in this study were deposited in the NCBI database, BioProject ID PRJNA736448. The data presented in this study are available on request from the corresponding author.

**Acknowledgments:** Portions of this research were conducted with high performance computing (HPC) resources provided by Louisiana State University. We thank Yuwu Chen and Le Yan for technical support in the HPC.

**Conflicts of Interest:** The authors declare no conflict of interest.

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
