# Peer review of "Changes in the Microbial Community in Soybean Plots Treated with Biochar and Poultry Litter"

_agronomy, doi:10.3390/agronomy11071428_

Round 1
Reviewer 1 Report
In the attached pdf document you will find detailed comments (please consider they are meant to be constructive).

Author Response
Our responses to reviewer's comments are in the attached word file.

Reviewer 2 Report
Manuscript ,,Changes of the microbial community in soybean plots treated with biochar and poultry litter,, contains many interesting results. It is also well structured. However, the handwriting still needs to be improved. A major revision is required.
My advice, recommendations and comments are listed below.
Please be sure that your manuscript thoroughly establishes how this work is fundamentally novel. Specific comparisons should be made to previously published materials that have a similar purpose. Please present a strong case for how this work is a major advance. This needs to be done in the manuscript itself, not just in the response to review comments. This is a very important point in terms of which I will further consider the manuscript.
Please be sure that your abstract and your Conclusions section not only summarize the key findings of your work but also explain the specific ways in which this work fundamentally advances the field relative to prior literature.
Introduction must be expanded and improved. The significance of this study should be more emphasize in the introduction. Take a look at this paper that may help you. https://www.mdpi.com/2071-1050/10/7/2536
Line 52: You can specify more specifically what pH and CEC values.
Line 35-37: This issue has been addressed in great detail by this very important paper and therefore the authors are encouraged to add it here as a reference. https://www.sciencedirect.com/science/article/pii/S0160412021001525
A list of the most important abbreviations should be given at the beginning or end of the manuscript so that the manuscript can be better oriented.
Line 92: ,,ha-1,, change to ,,ha-1,,
Line 112: In the entire manuscript, indicate the country of origin of all devices on which the experiments were performed.
Line 153: There are some typos in the manuscript, please check the text. It happens when writing.
Line 236: You could indicate measurement deviations. Did you also perform parallel analyzes?
Line 270: Enlarge figure 1, the descriptions are faintly visible.
Line 317: The same is true for Figure 2.
Line 355: It is interesting how the results differ in some cases from 2016 to 2017. What do you attribute it to?
Line 373: Consider adding some tables to the supporting information. The manuscript is very extensive, which does not change its good quality.
Line 399: I can't judge the results from figure 4 at all. It is very small and poorly visible. Please edit this so that readers can see the results well if published.
Line 539: Indicate the possible risks of such research. Add your recommendations for future research.
Line 585: Make sure the references are added correctly according to the journal's instructions.
Author Response

(The authors gave the same response as above.)

Round 2
Reviewer 2 Report
The manuscript has been significantly improved and thereforefore can be accept in its current form.